# Effect of Bis (2-Aminoethyl) Adipamide/Adipic Acid Segment on Polyamide 6: Crystallization Kinetics Study

**DOI:** 10.3390/polym12051067

**Published:** 2020-05-06

**Authors:** Yu-Hao Chen, Palraj Ranganathan, Chin-Wen Chen, Yi-Huan Lee, Syang-Peng Rwei

**Affiliations:** Institute of Organic and Polymeric Materials, Research and Development Center of Smart Textile Technology, National Taipei University of Technology No. 1, Sec. 3, Chung-Hsiao E. Rd., Taipei City 10608, Taiwan; mar820214@gmail.com (Y.-H.C.); rangapalraj@gmail.com (P.R.); cwchen@ntut.edu.tw (C.-W.C.); yihuanlee@mail.ntut.edu.tw (Y.-H.L.)

**Keywords:** polyamide 6, bis (2-aminoethyl) adipamide, copolyamide, isothermal crystallization kinetics, crystallization behavior

## Abstract

The crystallization behavior of novel polyamide 6 (PA6) copolyamides with different amounts of bis (2-aminoethyl) adipamide/adipic acid (BAEA/AA) segment was investigated. The wide-angle X-ray diffraction (WAXD) results showed that as the amount of BAEA/AA segment increased to 10 mole%, the crystalline forms of all PA6 copolyamide were transferred from the stable α-form to the unstable γ-form because of the complex polymer structure. According to studies of crystallization kinetics, the Avrami exponent (*n*) values for all copolyamide samples ranged from 1.43 to 3.67 under isothermal conditions, implying that the crystallization is involved in the two- to three-dimensional growth at a high temperature of isothermal condition. The copolyamides provided a slower crystallization rate and higher crystallization activation energy (Δ*E*_a_) than neat PA6. Polyamide containing 10 mole% of BEAE/AA content exhibited a unique crystallization behavior in the coexistence of the α and γ forms. These results deepen our understanding of the relationship between BAEA/AA content, crystal structure, and its crystallization behavior in low-melting PA6, and they make these types of copolyamides useful for their practical application.

## 1. Introduction

Nylons (namely, polyamides or PA) are one of the most vital high-performance commercial engineering thermoplastics because they offer attractive properties such as superior wear resistance, high toughness, excellent chemical and alkali resistance, and oil-proof qualities [1]. Of these polyamides, PA6 and PA66 are renowned commercial semi-crystalline engineering materials with excellent properties. The mechanisms of the crystallization of PA6 and PA66 have been thoroughly investigated to date, and their effects on the future performance of materials have been the subjects of considerable scientific research [2,3,4]. Although these vital polymeric materials have a very rapid crystallization rate, varying crystal structure, and morphology, some problems such as severe mold shrinkage and dimensional instability during the melting process may occur; these problems can adversely affect the final material properties [4,5,6]. Therefore, the evaluation of crystallization behavior to optimize processing conditions is very beneficial for the control of stability and properties and the development of final products [7]. The Avrami model is a widely used standard method for analyzing the crystallization behavior of semi-crystalline polymers [8].

Conventional aliphatic polyamides with more amide functional groups and high chain regularity result in higher melting temperatures, rapid crystallization rates, and higher crystallinity, which limit their applications. Currently, polyamides with low melting temperatures have received considerable attention due to their superior physical properties and are potentially used worldwide in foaming materials, hot melt adhesives, fusible yarns, and other products [9,10,11]. Several modification approaches have been used to improve the properties of polyamides; copolymerization is often used to reduce the melting temperature of polyamides. However, the crystallization behavior of the polyamide matrix is strongly affected by this approach. Hybart et al. [12,13] examined the crystalline properties of PA6 copolyamides containing the PA11 and PA66 segments and found that adding PA11 and PA66 segments into the PA6 backbone reduced the crystallization temperature, melting temperature, density, and volume contraction, which depending on the supercooling rate. Zhou et al. [11,14] reported that increasing the molar fraction of the PA510 segment in the PA6/66 backbone reduced the crystallization rate of the copolyamides. This copolyamide was successfully used as a hot glue for a metal sheet with low surface energy coating, thus improving the adhesive properties of the metal plate. Our research group [15] investigated the effect of PA6T segments on the crystallization kinetics of isomorphous PA66/6T copolyamides and non-isomorphous PA6/6T copolyamides. 

In our recent work, we synthesized a series of PA6 copolymers using a new bis (2-aminoethyl) adipamide (BAEA) diamine monomer. BAEA has a significant effect on destroying the crystallinity of PA6, and its copolyamides have outstanding mechanical properties for hot melt adhesive applications [16,17]. Additionally, a series of low-melting PA6 copolyamide has already been successfully synthesized successfully using other novel diamines (i.e., bis(4-aminobutyl) terephthalamide (BABT), bis (4-aminobutyl) adipamide (BABA), bis-(2-aminoethyl)adipamide (BAEA)) and organic salts (i.e., bis(4-aminobutyl) terephthalamide/sebacic acid (BABT/SA), bis (4-aminobutyl) adipamide/sebacic acid (BABA/SA), bis (4-aminobutyl) adipamide/adipic acid (BABA/AA), bis (2-aminoethyl) adipamide/1,4-cyclohexanedicarboxylic acid (BAEA/CHDA)), and bis (2-aminoethyl) adipamide/adipic acid (BAEA/AA) prepared by our research group [17,18,19,20,21]. On the other hand, the effect of BABA/SA on the crystallization behavior in the main chain of PA6 was also described using the method of isothermal crystallization kinetics [22]. Herein, the effect of BAEA/AA segments on the crystallization and melting behavior of PA6 in copolyamides was further investigated using the isothermal crystallization method, wide-angle X-ray diffraction, and polarized optical microscopy analysis. Those studies are necessary and meaningful for the actual application of these copolymers.

## 2. Experiment

### 2.1. Materials

In this work, PA6 copolymers (Scheme 1) had different bis (2-aminoethyl) adipamide/adipic acid (BAEA/AA) contents at molar ratios of 0, 5, 10, and 15 mole%, respectively. All of the required materials were used as received, ε-caprolactam (CPL, 99%) was purchased from the China Petrochemical Development Co. (Kaohsiung, Taiwan), the adipic acid (AA, 99.8%) was supplied by the Asahi Kasei Corporation (Tokyo, Japan). The BAEA monomer was synthesized from dimethyl adipate (DMA) and ethylenediamine (EDA), as previously reported [17,21]. The copolyamide composition of the PA6 and BAEA/AA segments are summarized in Appendix A, and the analysis of the chemical structure and thermal properties are listed in Appendix A, Appendix A and Appendix A. All the samples were dried in a vacuum oven before the experiments.

### 2.2. Isothermal Crystallization Kinetics Procedures

The isothermal crystallization kinetics experiment in this study involved a differential scanning calorimeter (DSC, PerkinElmer DSC8000, Waltham, MA, USA) in a nitrogen-rich atmosphere. PA6 copolyamide samples were heated to 250 °C at a rate of 10 °C min^−1^. This temperature was kept for 5 min to abolish the thermal history. Subsequently, the samples were quenched to specific crystallization temperatures at a rapid cooling rate of 100 °C min^−1^ and maintained at this temperature for 20 min until crystallization was complete. Both exothermic and endothermic DSC curves were recorded and analyzed.

### 2.3. Polarized Optical Microscopy (POM)

POM observations were performed using a Nikon Eclipse LV100N POL (Tokyo, Japan) microscope system equipped with a hot stage. All samples were melted at 250 °C between two glass slides to obtain samples for POM testing, and this temperature was maintained for 5 min to abolish the thermal history. Subsequently, the temperature was first increased and then rapidly reduced to the specific crystallization temperature ranges to observe the crystal formation. The observation conditions were related to the results of DSC measurements.

### 2.4. Wide-Angle X-ray Diffraction (WAXD)

The crystal structure of the polyamides was analyzed using an X-ray diffractometer (X’Pert³ Powder, Malvern, UK) equipped with copper Kα radiation (*λ* = 0.1542 nm, *V* = 40 kV). The scanning range for WXRD was from 2*θ* = 10–40°, and the step size was 0.013°. All the polymers were re-crystallized on a hot stage that was cooled at a target temperature for 20 min.

## 3. Results and Discussion

### 3.1. The Crystal Structure of PA6 and Its Copolyamides

PA6, PA6-5, PA6-10, and PA6-15 were prepared from the isothermal process at specific temperatures of 200, 155, 115, and 155 °C to understand the crystallization behavior of PA6 and PA6 copolyamides. WAXD analysis illustrates the influence of different content of the BAEA/AA segments on the crystalline transformation of PA6. Figure 1 shows the WAXD results of PA6 and PA6 copolymers. As displayed in Figure 1a,b, it is evident that the crystalline form of pure PA6 and PA6-5 is consistent and is mainly α-phase crystal with two diffraction peaks at about 2*θ* = 20° (200) and 23° (002, 202), indicating they have the same crystalline forms. Interestingly, when the amount of BAEA/AA reached 10 mole%, the copolyamide presented the coexistence crystal of α_2_ form (002, 202) and γ form (2*θ* = 21° (100), Figure 1c), and when BAEA/AA reached 15 mole%, the PA6-15 copolyamide presented a γ form (Figure 1d) [15,23]. These results demonstrated that the addition of the BAEA/AA segment induced the transformation of α form to γ form; the copolyamides formed a crystal structure dominated by γ form due to a change in the ordered structure of the PA6 backbone. It is well known that the α form of PA6 is more thermodynamically stable than the γ form [23,24,25]. In this study, the crystalline structure of PA6 (BAEA/AA) copolyamides was unfavorable to form a stable α form. Dasgupta et al. [23] reported that the crystal transformation of PA6 involved the relationship between the interaction of hydrogen bonds and the role of methylene packing. The addition of BAEA/AA segments restricted the ordered arrangement of hydrogen-bonded of the α form, thus promoting the formation of γ crystal in the PA6 copolymer, and the better packing of the methylene chains can make γ form more stable [23,26,27]. Furthermore, the crystallite size (*L*_HKL_) was confirmed using the Scherrer equation (Equation (1)) to more clearly explain the relationship between the crystallization behavior and the crystal size of the copolyamides [28,29,30,31,32].
(1)$$L_{\mathrm{HKL}}=\frac{K\;\lambda }{\beta \;\cos\theta }$$
where *K* is the shape factor, which adopts a value of 0.89 of all peaks, and *β* is the full-width at half maximum (FWHM) of the diffraction peak in radians (i.e., FWHM × π/180). The average crystal sizes of all copolyamides decreased from 7.96 to 4.87 nm, with the increased molar ratio of BAEA/AA, and the detailed crystallite parameters are listed in Table 1.

### 3.2. The Melting Behavior by Equilibrium Temperature

The isothermal thermal analysis of PA6 and its copolyamides was carried out by cooling the fused polymers to a specific crystallization temperature (*T*_c_). The Hoffman–Weeks equation was used to confirm the equilibrium melting point (*T*_m_^o^) of PA6 copolyamides, which is obtained by the linear equation extrapolation intersecting the line of *T*_m_ = *T*_c_. The equation is expressed as follows [33]:(2)$$T_{m}=T_{m}^{o}\left(1\;-\;\frac{1}{\gamma }\right)+T_{c}(\frac{1}{\gamma })$$
where *T*_m_ was obtained through the experiment, *T*_c_ is the crystallization temperature at the different isothermal processes, and *γ* is the thickening coefficient with polymers that depended on the crystal type and size [15].

Figure 2 shows the heating DSC traces of pure PA6 and PA6 copolyamides after isothermal crystallization at specific temperatures, and their details are listed in Appendix A. The Figure 2c reveals triple melting peaks in PA6-10, where peak I and peak II are the melting peak of the two different crystal forms, caused by BAEA/AA within the PA6 backbone, and peak III is the typical recrystallization performance. In contrast, a single *T*_m_ was found in PA6-15 (Figure 2d). All of the *T*_m_^o^ values were determined through Equation (2) [34]. The *T*_m_^o^ values of copolyamide decreased drastically from 259.1 to 196.7 °C, the *T*_m_^o^ for PA6, PA6-5, and PA6-10 were determined by peak I as 259.1, 255.5, and 197.6 ° C, respectively, and the *T*_m_^o^ for PA6-10 and PA6-15 were determined by peak II as 173.4 and 196.7 °C. The *T*_m_^o^ values of the copolyamide decreased sharply from 259.1 to 196.7 °C, indicating that the BAEA/AA segment may restrict the crystal growth of the PA6 backbone to form a tapering in crystal thickness, which is consistent with the WAXD results [35,36,37,38]. It is noteworthy that two different dependencies between *T*_c_ and *T*_m_ (peak I and peak II corresponding to Figure 3a,b) were observed using the Hoffman–Weeks method, which implied that the crystal growth of peaks I and II corresponded to the thermodynamically stable α-form and unstable γ-form, respectively [25,35,39,40,41,42]. 

### 3.3. Isothermal Crystallization Kinetics

A series of PA6 copolyamides were evaluated using the isothermal crystallization method to confirm the retarding effect of the BAEA/AA segment on the ordered stacking of the PA6 backbone. The nucleation growth of all the polyamide samples is investigated in the following isothermal process ranges: 180–200 °C, 135–155 °C, 95–115 °C, and 135–155 °C, respectively. The relative crystallinity of the copolyamides (*X*(*t*)) is evaluated as a function of time, as follows:(3)$$X(t)=\frac{X_{c}\left(t\right)}{X_{c}(t=\infty )}=\frac{\int_{0}^{t}\;\frac{{dH}_{c}\left(t\right)}{dt}\;dt}{\int_{0}^{t=\infty }\;\frac{{dH}_{c}\left(t\right)}{dt}\;dt}$$
where *X*(*t*) and *X*_c_(*t* = ∞) are the relative crystallinity at the time (*t*) and infinite time (*t* = ∞), respectively, and d*H*_c_(*t*) is the enthalpy of crystallization. Figure 4 plots *X*(*t*) as a function of the time for pure PA6 and a series of PA6 copolyamides at given isothermal crystallization temperatures. The crystallinity isotherm curves of all polyamide specimens have a “sigmoidal” shape depending on the time and shift to the right with the increasing isothermal crystallization temperatures, representing a gradually decelerating crystallization rate. 

Presuming that the (*X*(*t*)) of the copolyamides increases with escalating crystallization time, we applied the Avrami equation [15,43,44] to examine the crystallization kinetics process:(4)$$X(t)=1\;-\;\exp(-{Kt}^{n})$$
where *K* is the crystallization rate, and n is the Avrami constant with a value depending on the type of nucleation and crystal growth dimension. Equation (4) can be further expanded employing double natural logarithms to acquire the crystallization parameters, as follows:(5)ln(−ln (1−X(t))=n ln(t)+lnK

The Avrami parameters, n, and rate constant value, *K*, for PA6, and its copolyamides were acquired from the plots of ln[−ln(1−*X*(*t*))] curves as a function of ln(*t*) at various isothermal temperature process. Appendix A illustrates the Avrami plots of PA6 and its copolyamides at different isothermal crystallization temperature. A linear equation was employed to discover the K and n values, and the results are summarized in Table 2. The Avrami exponent (*n*) represents the relationship between the dimension of crystal growth and the crystallization nucleation mechanism. The Avrami parameter of *n* = 1–2 indicates that the dimension of the crystal growth proceeds in a one-dimensional manner (acicular shape); furthermore, *n* = 2–3 represents two-dimensional growth (disk shape), and *n* = 3–4 represents three-dimensional growth (spherulite shape) [1,45]. The crystallization half-life (*t_0.5_*) is an important parameter in the study of crystallization kinetics. The *t_0.5_* is the time at which 50% of the relative crystallinity is acquired. The *t_0.5_* was also used to elucidate the growth mechanism of crystallization, and its reciprocal can be defined as the growth rate of crystallization (*G*). The value of *t*_0.5_ was calculated from the equation between *n* and the *K*, which is expressed as follows:(6)$$t_{0.5}={\left(\frac{\ln\left(2\right)}{K}\right)}^{\frac{1}{n}}$$

From Table 2, the values of Avrami parameters, *n*, evidently increased with the increasing isothermal crystallization temperatures. The Avrami parameter (*n*) for PA6 = 1.43–3.27, PA6-5 = 1.76–3.67, PA6-10 = 1.76–2.90, and PA6-15 = 1.68–2.57, which means that the nucleating mechanism of the PA6 and its copolyamides chains transformed from one-dimensional to two- or three-dimensional growth as the isothermal temperature increased. With the addition of BAEA/AA 5 mole%, the n values ranged from 1.76 to 3.67, which means that the addition of the BAEA/AA comonomer segment marginally influenced the nucleation mechanism and the growth of the PA6 crystallites. The aforementioned result exposes that the isothermal crystallization kinetics of the PA6-5 may transform into a mixture mode of two-dimensional and three-dimensional growth space extension due to the spherulitic crowding and impingement. From Table 2, it can also be witnessed that n values reduce as the mole% of BAEA/AA in the PA6 rises, which confirms that there is a rise in the heterogeneity during the growth step owing to the restricted mobility of the PA6 copolyamide. Besides, the small Avrami exponent value, *n*, might be due to the faster crystallization growing process, which does not permit adequate time for growth in three-dimensions [46,47]. Furthermore, the three-dimensional crystal was difficult to constitute with the increasing BAEA/AA content, as demonstrated by the narrowing *n* value. Theoretically, the value of Avrami exponent *n* is an integer ranging from 1 to 4. However, the nucleation crystallization of the growth process involves homogeneous and heterogeneous nucleation at the same time, which causes the value of *n* to be not an integer. The *K* values are also listed in Table 2. The *K* values were greater at lower isothermal crystallization temperatures than higher isothermal crystallization temperatures for all copolyamides. Since *K* is the unit of min^−1^, it is not reasonable to apply *K* values to directly compare the crystal dimension of the diverse samples at different isothermal processes, if *n* values are not identical [48]. Based on the fact, it is difficult to compare the influence of varying BAEA/AA content on the crystallization behavior of PA6 in all samples. Hence, in the case of PA6-5 and PA6-15, the isothermal crystallization behavior of copolyamides with different BAEA/AA content was studied in the same target temperature range (see Table 2). The crystallization rate K and Avrami exponent *n* in PA6-15 were significantly lower than that of PA6-5, which reveals that BAEA/AA played a significant role in the crystallization rate and the mechanism of crystallization growth in the PA6 polymer chain. A small amount of the BAEA/AA within the PA6 backbone played an essential role in disrupting the ordered crystal arrangement [42]. In addition, the Avrami exponent *n* of PA6-5 exhibited superior value when compared to neat PA6, although its crystallization rate *K* was significantly lower, which was possibly due to the smaller amount of BAEA/AA segments, which enhances the softness of the overall PA6 structure and facilitating the formation of larger crystal dimension [22]. Interestingly, PA6-10 had a very low crystallization rate *K* and weak Avrami exponent *n*, which suggests that the coexisting crystal behavior of α and γ forms increased the difficulty of crystallization growth.

The plot of *G* versus the supercooling temperature (Δ*T* =*T*_m_^o^ − *T*_c_) is exhibited in Figure 5 to compare the crystallization behavior of the polyamides more clearly. The overall crystallization rate of all polyamides increases with increasing thermodynamic driving force (i.e., increased supercooling). In addition, the crystallization rate of two different driving forces was observed between the dominant crystallization behavior of α and γ forms, and this might be related to the different nucleations-controlled process in α and γ forms. 

### 3.4. The Activation Energy of Isothermal Crystallization

The activation energy of polymer crystallization includes the activation energy of chain movements and nucleation. The isothermal crystallization process for the original PA6 and its copolyamides were presumed to be thermally activated. The Avrami exponents *K* and *n* were applied to estimate the activation energy values for crystallization using the Arrhenius equation [15,49]:(7)$$K^{\frac{1}{n}}=k_{0}\;\exp\left(\frac{-\Delta E_{a}}{{RT}_{c}}\right)$$
where *k*_0_ is the temperature-independent pre-exponential factor, Δ*E*_a_ is the activation energy of crystallization in units of kJ mol^−1^, and *R* is the ideal gas constant (8.314 J K^−1^ mol^−1^). Δ*E*_a_ was determined from the slope of the linear plots of ln(*K*)/n versus 1/*T*_c_ and it is shown in Appendix A and Table 3.

The Δ*E*_a_ values of the original PA6 and PA6 copolyamides were −240.61, −135.42, −49.17, and −70.57 kJ mol^−1^, respectively. The value of Δ*E*_a_ was negative because the polymer released energy when changing from a molten fluid to a crystalline state [50]. The Δ*E*_a_ value increases with the mole% of BAEA/AA segments because the incorporation of the comonomer within the PA6 matrix disturbs the chain regularity and decreases the crystallization order. However, the Δ*E*_a_ of the PA6-15 at a value of −70.57 kJ mol^−1^ was lower than that of the PA6-10 in −49.17kJ mol^−1^. This is associated with the growth of crystallization, which is transformed from coexisting crystal into uniform crystal [14,15,51].

### 3.5. Spherulitic Growth for Lauritzen–Hoffman Equation

The Lauritzen–Hoffman equation was used to explain the spherulitic growth rate during the isothermal crystallization process of pure PA6 and its copolyamides. The Lauritzen–Hoffman equation assumes that the formation of lamellae and folding of polymer chains is the dynamic control and proposes a transfer mechanism of the system, demonstrating that the transfer of crystal growth is based on the secondary nucleation theory [14,15,51,52].
(8)$$G=G_{0}\exp\left(\frac{-\Delta U^{*}}{R(T_{c}-T_{\infty })}\right)\exp(\frac{-K_{g}}{T_{c}\Delta Tf})$$
where *G*_0_ is the pre-exponential factor, *R* is the ideal gas constant, *T*_c_ is the crystallization temperature, *f* = 2*T*_c_/(*T*_m_^o^ + *T*_c_) is the correction factor, ∆*T* = *T*_m_^o^ − *T*_c_ is the supercooling temperature, and *T*_∞_ = *T*_g_ − 30 is the temperature below which the molecular chain ceases motion. *K*_g_ is the nucleation constant related to the surface free energy of the lamellae, and ∆*U*^*^ is the activation energy necessary for the macromolecules to diffuse to the crystal phase in the melting state, which was obtained by the Williams–Landel–Ferry (WLF) equation. The equation and its parameters are as follows [53,54]:(9)$$\Delta U^{*}=\frac{C_{1}T_{c}}{C_{2}+T_{c}-T_{g}}$$
where *C*_1_ = 4120 cal mol^−1^, *C*_2_ = 51.6 K. The crystal growth rate of *G* at each crystallization temperature can be obtained from the reciprocal of *t*_0.5_; Appendix A plots the ln*G* + Δ*U*^*^/[R(*T*_c_ − *T*_∞_)] versus 1/(*T*_c_Δ*Tf*) for all the polyamides, and the values of *K*_g_ and *G*_0_ can be acquired from the slope and intercept of the linear equation, respectively. The values of *K*_g_ for the polyamides are summarized in Table 3. The curves of the polyamides had good linear relationships, which demonstrates that the Lauritzen–Hoffman method can be used to analyze the crystallization growth of each polyamide [51]. Evidently, copolyamides with high BEAE/AA segment have a notable increase in *K*_g_, which is related to the increase of the presence of structural irregularities in the polymer chain and corresponds to a more difficult and complicated crystallization nucleation process [55,56]. Interestingly, the *K*_g_ value weakened in PA6-15, which was caused by a more uniform crystal growth process. 

### 3.6. Polarized Optical Microscopy Observation 

POM images of neat PA6 and its copolyamides are shown in Figure 6**.** In order to observe and compare the morphology of the PA6 and its copolyamides clearly during the isothermal process, they were supercooled at a specific temperature for 7 min (which guaranteed that their crystal growth was completed). As shown in Figure 6, a crystal pattern was detected under the isothermal process, the neat PA6 tend to form sheet crystals with a larger size and high packing density, and the copolyamides tended to form smaller spherical (three-dimensional) or discoid (two-dimensional) crystals [15,22,57,58,59,60,61]. Considering the result of crystallization kinetics, it is evident that the copolyamides preferred to exist in the crystalline form of two dimensions. In addition, the crystal size of PA-10 was less than that of neat PA6, indicating that a large amount of nucleus was generated in a short period of time and restrained the continuing growth of the large crystal. This happened because the BAEA/AA segments ruined the symmetry and regularity of the polymer chains for the PA6 copolyamides and raised its intra-crystalline defect. A similar result was found in the crystallization behavior of most of the copolymers (i.e., the crystal size reduced), which indicates that the PA6 with the BAEA/AA structure was non-isomorphous [15,22,42,62].

## 4. Conclusions

In this work, the crystallization and melting behavior of pure PA6 and its copolyamides were evaluated at given crystallization temperatures and cooling rates. According to the WAXD and melting behavior results, with the increase of BAEA/AA content, the transformations of crystal forms from α to γ form that was more thermodynamically unstable and found the tapering crystal size. Isothermal crystallization kinetics demonstrated that the crystallization rate and crystallization dimension were restricted by the loading of the BAEA/AA segment on the PA6 backbone. The Avrami exponent n was in the range of 1.43–3.67 for the isothermal condition. The crystallization half-life of copolyamides was higher than that of pure PA6, which implies that the crystallization rate reduced when the BAEA/AA segment was loaded into the PA6 backbone. Additionally, the Δ*E*_a_ of crystallization was in the values of −240.61, −135.42, −49.17, and −70.57 kJ mole^−1^, and the *K*_g_ was in the values of 1.16 × 10^6^, 4.71 × 10^6^, 5.67 × 10^6^, and 6.59 × 10^5^ K^2^, which correspond to PA6, PA6-5, PA6-10, and PA6-15, respectively. The above results indicated that the highest values of Δ*E*_a_ and *K*_g_ were found in PA6-10, because the BAEA/AA segment destroyed the symmetry and regularity of PA6 chains and restricted the folding of PA6 chains into the nucleus. In addition, the crystal growth process of PA6-10 involved α and γ-competitive forms, which meant that crystal growth became more difficult and complicated. Based on our study, these findings deepened our understanding of this series of low-melting copolyamide 6, and its properties and were beneficial to its practical applications soon.

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
