# Peer review of "Effect of Bis (2-Aminoethyl) Adipamide/Adipic Acid Segment on Polyamide 6: Crystallization Kinetics Study"

_polymers, 2020, doi:10.3390/polym12051067_

Round 1

Reviewer 1 Report

The paper about the effect of bis (2-aminoethyl) adipamide/adipic acid (BAEA/AA) segment on the PA6 copolymer is well written and can be published after minor revision. In the following, some suggestions.

There are some typos in the introduction.

Define all the parameters and constants adopted in the equations. Pay attention to the equation 2, this referee suggests adopting another symbol for the thickening coefficient, furthermore, what value was adopted for this coefficient.

Try to align the plots of Figure 3.

The authors should deepen the discussion on the POM shown in figure 6, some POM with higher magnification would help to highlight the differences between the neat PA6 and the copolymers.

Reviewer 2 Report

The authors made several major changes on the route toward improving the manuscript. However, there are still some problems which should be corrected  before considering it for publication. The most important issued which should be changed are related to the WAXD results:

- the authors changed the title of the paragraph 3.1 to "The Crystal conformation of PA6 and its Copolyamides”. This title is incorrect in physical terms because there is no conformation of crystals. There is conformation of molecules but not of crystals. In this case the authors can write "Crystal structure" or "Crystal forms". The term "crystal conformation" is used consequently by the authors in the revised version, e.g. row 154 or 308-309. It should be changed in all places to crystal forms or crystal modifications;

- the legend for Fig. 1 c is incorrect. According to the data in tab. 1, the crystals peaks are from γ and α2 forms;

- there is still a problem with language. For instance in rows 97-93 there is sentence "The observation conditions were associated with the results of DSC measurements." The word "associated with" is incorrect in this context and should be replaced by "related to" or correlated with". In rows 142-143 there is sentence "....thickening coefficient with polymers with respect to the crystal type and size." with bad using of prepositions. The whole text should be again carefully read and the language needs still improving

Author Response

This manuscript is a resubmission of an earlier submission. The following is a list of the peer review reports and author responses from that submission.

Round 1

Reviewer 1 Report

The paper ''Influence of Bis (2-Aminoethyl) Adipamide/Adipi Acid Segment within Polyamide 6: Isothermal a Non-isothermal Approach'' considers the influence of BAEA/AA segments within PA on the crystallization kinetics. They found a significant influence of the amount of BAEA/AA on the melting temperature, activation energy, n and Kg. The authors conducted a good experimental investigation, however, they need to expand the discussion on the mechanisms that induce the mentioned behavior during the crystallization. Furthermore, in the introduction, the authors mention that the presence of BAEA/AA segments within PA could improve the properties of final polymeric parts, however, they did not demonstrate this statement. This referee suggests analyzing the mechanical behavior of the parts made of the copolymer proposed in this work (for instance, films produced by compression molding and analyzed by DMA). For all these reasons, major review is recommended. 

In the following, other suggestions:

The authors should pay attention to the acronyms, some of them was written with wrong typing.

line 134-138: what figure the authors refer to when they mention the sharp decrease of Tm0 and the drastic increase of γ? 

equation 2: what are the values adopted for lamellar thickness, surface free energy and the other parameters adopted in eq 2?

line 174-179: it would be recommended a morphological investigation of the crystal forms by SEM of AFM?

Figure 4: the POMs do not show the crystals that the authors describe. It is recommended enlargement of the micrographs or to adopt a different analysis.

line 283: pay attention to the font

Table 5: what is Kg? the authors should define it in the table caption.

ref 16: what are the journal, the volume and the pages of this reference?

references: 50% of the cited literature was written before 2003. The authors should consider more recent literature.

Conclusions: The authors could try to resume the mechanisms that regulate the behavior during crystallization in a critical manner. 

Reviewer 2 Report

The problems with the manuscript "Influence of Bis (2-Aminoethyl) Adipamide/Adipic Acid Segment within Polyamide 6: Isothermal and Non-isothermal Approach" by Yu-Hao Chen , Palraj Ranganathan , Chin-Wen Chen , Yi-Huan Lee , Syang-Peng Rwei starts with the very beginning, namely from the title. According to the title, the manuscript  is devoted to the influence of bis (2-aminoethyl) adipamide/adipic acid segments within PA-6. There is lack in logical  structure of this title - it should be explicite written that this is influence on something, for instance kinetics of crystallization. May be this lack in the title is a simple result of general problem of this manuscript. The authors performed a lot of various experiments and applied various procedures for data interpretation trying to describe the effects of bis (2-aminoethyl) adipamide/adipic acid segments on various aspects of PA-6 crystallization. Due to too large number of experiments, the nonisothermal part, which is announced in the title, was inserted in the supplementary information, making unacceptable assymetry in the information between isothermal and nonisothermal parts in the main text. There is total informational chaos in data interpretation increased additionally by misinterpretation of some of the results as well as by terrible English language.  If the authors would like to send this manuscript somewhere else, the reviewer strongly suggest to consider choosing most important procedures for data interpretation and provide the results with reasonable, as deep as possible, physical interpreation. Additionally, there is strong need to make statistical analysis for evaluation of data significance.  Some of the main problems, of course not exhausting all of them, are listed below:

  • in row 134 it is written, that the chains form "complete and stable lammelar structure". There is no confirmation in the results that this structure is complete and stable. 
  • the sentence in rows 140-142  does not make sense
  • row 151 - what is the meaning of the terms "sensitive slope" and "gentle slope" ?
  • in rows 162-165 the authors write that the diffraction peaks correspond to crystal morphology of the alpha form. It is well known that the morphology of crystals refers to the outer form of the crystal, for instance, needle morpholgy, and WAXD peaks have nothing to do with morphology of crystals
  • there is a lot of problems with WAXD analysis on page 7. The authors do not provide readers with numerical deconvolution of WAXD profiles which would result in clear angular positions of particular peaks.  Without such deconvolution there is only doubt if the shift in position of alpha2 peak is true or this is artefact. By the way, if such shift is true it needs comments
  • there is lack of some essential interpretations, for instance why the Avrami exponent "n" increases with crystallization temperature ?
  • the authors determined the activation energy as well as nucleation constant Kg using some equations which need for such determination the linear dependencies of the results presented  in the appropriate coordinates system. It is evident that the results in Figs. 8 and 9 are not linear. So, both determinations are not allowed on the basis of presented results
  • in conclusions it is written that the exponent "n" in nonisothermal regime is higher because of heterogeneous nucleation. This is completely misinterpretation because similar heterogenous nucleation is present during homogenous nucleation. There are others mechanisms which are not discussed by the authors